# Dopamine-Conjugated Carbon Dots Inhibit Human Calcitonin Fibrillation

**DOI:** 10.3390/nano11092242

**Published:** 2021-08-30

**Authors:** Jhe-An Wu, Yu-Chieh Chen, Ling-Hsien Tu

**Affiliations:** Department of Chemistry, National Taiwan Normal University, Taipei 106, Taiwan; xxxddd519@gmail.com (J.-A.W.); jj40344030s@gmail.com (Y.-C.C.)

**Keywords:** human calcitonin, protein aggregation, amyloidosis, carbon dots, dopamine

## Abstract

The development of biocompatible nanomaterials has become a new trend in the treatment and prevention of human amyloidosis. Human calcitonin (hCT), a hormone peptide secreted from parafollicular cells, plays a major role in calcium–phosphorus metabolism. Moreover, it can be used in the treatment of osteoporosis and Paget’s disease. Unfortunately, it tends to form amyloid fibrils irreversibly in an aqueous solution, resulting in a reduction of its bioavailability and therapeutic activity. Salmon calcitonin is the replacement of hCT as a widely therapeutic agent due to its lower propensity in aggregation and better bioactivity. Herein, we used citric acid to synthesize carbon dots (CDs) and modified their surface properties by a variety of chemical conjugations to provide different functionalized CDs. It was found that dopamine-conjugated CDs can effectively inhibit hCT aggregation especially in the fibril growth phase and dissociate preformed hCT amyloids. Although the decomposition mechanism of dopamine-conjugated CDs is not clear, it seems to be specific to hCT amyloids. In addition, we also tested dopamine-conjugated mesoporous silica nanoparticles in preventing hCT fibrillization. They also can work as inhibitors but are much less effective than CDs. Our studies emphasized the importance of the size and surface functionalization of core materials in the development of nanomaterials as emerging treatments for amyloidosis. On the other hand, proper functionalized CDs would be useful in hCT formulation.

## 1. Introduction

Amyloid formation and accumulation have been shown to cause a wide range of diseases, including neurodegenerative diseases such as Alzheimer’s disease, Parkinson’s disease, and Huntington’s disease [1]. In addition, the aggregation of islet amyloid polypeptide (IAPP) and insulin is implicated in the pathogenesis of type 2 diabetes [2]. Human calcitonin (hCT), a 32-residue hormone peptide secreted from the parafollicular cells (also known as C-cells) of the thyroid gland, is also one of the amyloidogenic proteins [3,4]. The main physiological function of calcitonin is the regulation of calcium and phosphate concentration in the blood. It has an opposite role to parathyroid hormone and reduces calcium levels in the blood through two main mechanisms: by inhibiting the activity of osteoclasts, which are the cells responsible for the bone breakdown, and by decreasing the resorption of calcium in the kidneys. Therefore, the hormone calcitonin is also used in the treatment of osteoporosis, Paget’s disease of bone, and sometimes, for bone pains [5]. Patients with medullary carcinoma of the thyroid may develop localized amyloid deposition in the tumors, but the cause is still unknown [6]. Due to the aggregation propensity of hCT, it is currently replaced by salmon calcitonin as an active pharmaceutical ingredient (API) of the drug product [7,8]. Scientists are still seeking a suitable formulation to stabilize hCT or prevent hCT fibrillization.

So far, compared with other amyloidogenic proteins such as amyloid-beta (Aβ) and IAPP, there have been few reports suggesting effective inhibitors for hCT fibrillization. Epigallocatechin-3-gallate (EGCG), extracted from green tea, is the first small molecule characterized by nuclear magnetic resonance to inhibit hCT oligomerization. Tyr11, Phe16, and Phe22 of hCT were suggested to be involved in interaction with the aromatic ring of EGCG to block peptide assembly [9]. Later, two major polyphenol compounds, magnolol and honokiol, found in the Chinese herb *Magnolia officinalis,* were shown to inhibit both oligomerization and fibrillization of hCT. Data derived from the isothermal titration calorimetry experiment suggested that both Magnolol and honokinol can directly bind to hCT [10]. Cucurbit [7] uril is a macrocyclic molecule made of seven glycoluril monomers and can encapsulate aromatic residues [11]. Therefore, it was also utilized to block peptide assembly and prevent hCT fibrillization. More recently, a group of polyphenolic molecules were tested for their ability to inhibit hCT amyloid formation. The results suggested that the structural feature of the vicinal hydroxyl (catechol) group was essential for their ability in disrupting the process of fibril formation [12]. These studies may provide great help in drug design.

Water-soluble carbon dots (CDs), as a bio-friendly nanomaterial with a tiny size less than 10 nm, have recently been widely studied for potential biomedical applications, due to their low toxicity, high biocompatibility, and various functionalities [13,14]. Small-molecule-conjugated CDs with proper functional groups also allow nanoparticles to be used as chemosensors to detect enzyme activity or the presence of heavy metals [15,16,17,18]. Some functionalized CDs even could serve as an initiator for polymerization [19]. The synthetic approach of these CDs is usually simple and eco-friendly. In addition, CDs have been reported to be inhibitors of amyloid formation by Aβ, IAPP, and insulin [20,21,22]. Thus, a biocompatible nanoparticle-based approach could present a more promising alternative for inhibition of amyloid aggregation. In this study, we synthesized CDs through a bottom-up method, conjugated different functional groups to change its surface properties, and investigated the effects on the aggregation of hCT through several biophysical techniques to explore the potential applications of these materials in pharmaceuticals. Among them, it is worth noting that dopamine-conjugated carbon dots (DA-CDs) most effectively inhibited hCT amyloid formation. We speculated that π–π interaction would play a crucial role between hCT and DA-CDs. In addition, we tested the ability of DA-CDs in dissociating preformed amyloid fibrils derived from Aβ, IAPP, and hCT. It is surprising that DA-CDs could only decompose hCT amyloids. Moreover, we also prepared dopamine-conjugated mesoporous silica nanoparticles (DA-MSNs) with a larger core size. DA-MSNs could still inhibit hCT fibrillization in a concentration-dependent manner but were much less effective than DA-CDs, suggesting catechol groups strongly interact with hCT. Highly soluble DA-CDs might be a better choice to stabilize protein and prevent protein aggregation. This study is the first to use biocompatible nanoparticles in an hCT investigation. To discontinue using salmon calcitonin as API in bone disease treatment, we must find a better solution to stabilize hCT. This study points to the potential of using DA-CDs in hCT formulation.

## 2. Materials and Methods

### 2.1. Peptide Synthesis and Purification

hCT, IAPP, and Aβ42 peptides used in this study were synthesized through an automatic microwave-assisted solid-phase peptide synthesizer with fluorenylmethyloxycarbonyl (Fmoc) chemistry. In order to conform to the state of human calcitonin in the human body (amidated C-terminus), we used the Fmoc protected Rink Amide ProTide synthetic resin (0.18 mmole/g) as the solid support and used Cl-MPA ProTide Resin (0.16 mmole/g) for Aβ42 synthesis. First of all, the Fmoc group on the synthetic resin was removed by 20% piperidine in dimethylformamide (DMF). The exposed amine group on resin can couple with the activated carboxyl group of the first amino acid in which the amino group was also protected by Fmoc. N,N’-diisopropylcarbodiimide was used here as activation reagent. This reaction cycle was repeated until the completion of the synthesis. To improve synthesis yield, β-branched amino acids such as Thr, Ile, Val, and Pro executed a coupling reaction twice with its next amino acid. In addition, the temperature of reaction for His was set at 50 °C to reduce racemization. Later, the cleavage cocktail including trifluoroacetic acid and scavengers (triisopropylsilane, 3,6-dioxa-1,8-octanedithiol, H_2_O) was prepared with a ratio of 92.5:2.5:2.5:2.5 to cut peptide from the synthetic resin and protective groups from amino acid side chains. Crude peptides were precipitated in cold ether, filtered, and freeze-dried on a lyophilizer. Subsequently, a small amount of each crude sample was first purified by a reverse-phase high-performance liquid chromatography (RP-HPLC) and subjected to electrospray ionization mass spectrometer or matrix-assisted laser desorption ionization–time-of flight (MALDI-TOF) mass spectrometry to confirm the molecular weight. Next, crude peptides of hCT and IAPP were further dissolved in 50% acetic acid solution (1 mg/mL) and mixed with I_2_ in methanol to form the disulfide bond between two cysteine residues. Peptides were purified by RP-HPLC using a Proto 300 C18 semipreparative column. A two-solution gradient was used for hCT and IAPP purification: solution A consisted of 100% H_2_O and 0.045% HCl (*v*/*v*) and solution B consisted of 80% acetonitrile, 20% H_2_O, and 0.045% HCl. Another two-solution gradient was used for Aβ purification: solution A was composed of 100% H_2_O, 0.05% NH_4_OH, and 0.005% DEA (*v*/*v*) and solution B was composed of 90% acetonitrile, 10% H_2_O, 0.05% NH_4_OH, and 0.005% DEA (*v*/*v*). The collected fractions monitored by UV signal at 215 nm were pooled and lyophilized. The molecular weight of three peptides was confirmed by MALDI-TOF mass spectrometry (Bruker, MA, USA, ultrafleXtreme^TM^) again. The HPLC chromatogram and mass spectrum for hCT are shown in Appendix A.

### 2.2. Synthesis of CDs

Calcination of citric acid anhydrous placed in a glace petri dish took place in an oven at 180 °C for 2.5 h. The synthesized CDs were subsequently suspended in water and purified by dialysis (molecular weight cut-off: 1 kDa) for 24 h [23]. The deionized water was changed every one hour in the first 4 h. Finally, water was completely removed by lyophilizer. The synthesis yield was about 15% after dialysis. The prepared CDs were further modified to show different functionality. In general, 37.5 mg CDs with rich carboxyl groups were activated by reacting with 250 mg EDC·HCl and NHS in 10 mL DDI water and stirred for 1 h. An excessive amount of amine containing small molecules such as dopamine, ethanolamine, ethylenediamine, and β-alanine was added into solutions individually to produce DA-CDs, ETA-CDs, ED-CDs, and β-a CDs (Figure 1). After 24 h, the mixture was freeze-dried. Parts of crude CDs were further purified by dialysis (molecular weight cut-off: 1 kDa) for 24 h before experiments. DA-CDs were synthesized twice to ensure their inhibition in hCT fibrillization.

### 2.3. Synthesis of MSNs

The procedure for synthesis of MSNs followed the reference method with some modifications. First of all, 0.51 g hexadecyl trimethyl ammonium chloride (CTAC) and 72.5 μL triethanolamine (TEA) were dissolved in 20 mL DDI water and stirred at 95 °C for 1 h. Then, 1.5 mL tetraethyl orthosilicate (TEOS) was added at a flow rate of 0.5 mL/min into the mixture, which was maintained at 95 °C and stirred for another 1 h. The products were collected by centrifugation at 9000 rpm for 20 min and washed three times with ethanol to remove extra reagents. The collected products were extracted with 1% sodium chloride (NaCl) in methanol for 3 h at room temperature to remove the template CTAC. Later, 150 mg MSNs was first dispersed in 150 mL ethanol and sonicated for 10 min. The solution was refluxed for 4 h with the addition of 300 μL 3-aminopropyltrimthoxysilane (APTMS) to produce amine-functionalized MSNs (MSN-NH_2_). The resulting solution was centrifuged and washed with DDI water three times. Then, 20 mg MSN-NH_2_ was further reacted with 200 mg succinic anhydride in dehydrated DMF at room temperature for 5 h to produce carboxyl-functionalized MSNs (MSN-COOH). Finally, dopamine (11.7 mg) was covalently conjugated onto MSN-COOH (5 mg) by using cross-linking reagent EDC (6 mg) and NHS (3.6 mg) in 1 mL deoxygenated MES buffer at pH 5.0. Again, dopamine-conjugated MSN (DA-MSN) was washed with DDI water several times to remove the above reagents and finally, dispersed in water for storage.

### 2.4. Characterization of CDs and MSNs

The particle size and zeta potential of CDs and MSNs were measured by ELSZ-2000 (Otsuka Electronics, Osaka, Japan). The Fourier transform infrared spectra (FTIR) spectra were recorded to identify the functional groups of CDs and MSNs (Spectrum 500, PerkinElmer, Waltham, MA, USA). The elemental composition of CDs was measured by X-ray photoelectron spectroscopy (XPS, ESCALAB Xi^+^, Thermo Fisher Scientific, Waltham, MA, USA). The optical properties of CDs and MSNs were studied using a microplate reader (SpectraMax M2, Molecular Devices, Sunnyvale, CA, USA).

### 2.5. Peptide Preparation

About 0.1 mg peptide was first treated with 100 μL of hexafluoro-2-propanol (HFIP) for 5~6 h at room temperature to dissolve pre-aggregates and then lyophilized. The resulting Aβ peptide powder was further treated again with 100 μL, 3 mM NaOH solution. Dry hCT peptide powder was dissolved in 300 μL, 50 mM phosphate buffer (pH 7.4), but IAPP and Aβ were dissolved in 10 mM Tris buffer at pH 7.4 and pH 7.8, respectively. Peptide solutions were centrifuged at 15,000 rpm for 10 min to remove any insoluble aggregates. An amount of 10 μL of each peptide stock solution was used to determine protein concentration using BCA protein assay kit (Thermo Fisher Scientific, USA) according to user manuals, and later, the rest was diluted to the desired concentration by an appropriate buffer.

### 2.6. Lipid Vesicle Preparation

Lipids containing 1-palmitoyl-2-oleoyl-glycero-3-phosphocholine (POPC, 49%), 1-palmitoyl-2-oleoyl-sn-glycero-3-phospho-L-serine (POPS, 21%), and cholesterol (30%) were first dissolved in 100% chloroform in a 50 mL round-bottom flask. The chloroform was then evaporated with a stream of nitrogen gas to form lipid films, which were dried under vacuum for 2–3 h to completely remove the residual organic solvent. The lipid film was rehydrated by adding pH 7.4, 50 mM phosphate buffer. The solution was vortexed several times until the lipids were fully dispersed. The multilamellar vesicles were then subjected to 10 freeze−thaw cycles and extruded 11 times through double-stacked 100 nm pore size filters with an Avanti mini-extruder. The phospholipid concentration was determined using a phospholipid assay kit (colorimetric method, Abnova, Taipei, Taiwan).

### 2.7. Thioflavin-T (ThT) Assay

In this study, ThT assay was conducted in several different conditions to test the inhibitory effect of DA-CDs. In general, peptide solution was prepared in buffer with 16 μM ThT. CDs were prepared in DMSO. The final concentration of DMSO in solution was commonly fixed at 1%, even for control studies. Assays were performed at 25 °C in a sealed 384-well nonbinding surface microplate (Corning, NY, USA) with agitation every 2 h. Measurements were made using a multimode microplate reader (SpectraMax M2, Molecular devices, Sunnyvale, CA, USA) with excitation at 430 nm and emission at 45 nm. Data were collected once every 2 h, averaged, at least, from triplicate wells, and plotted as fluorescence versus time. For seeding experiments, the seeds were prepared by incubating 64 μM hCT in microtubes in a ThermoMixer (Eppendorf, Hamburg, Germany) with frequency shaking (1 min per 10 min) at 500 rpm for 3 days to form mature fibrils, and 10% seeds (in monomeric units) were added at the beginning of the reaction. For samples that were also subjected to dynamic light scattering (DLS) study, the same components but larger sample volumes were prepared in microtubes and incubated in a ThermoMixer. At desired time points, samples were taken out from microtubes for different measurements.

### 2.8. Transmission Electron Microscopy (TEM)

Five microliters of peptide solution from ThT assays was placed on a carbon-coated Formvar 300 mesh copper grid for 1 min and then negatively stained by incubation with 2% uranyl acetate for another 1 min. TEM was performed in an instrumentation center at National Taiwan University using a Hitachi H-7100 (Tokyo, Japan) transmission electron microscope with an accelerating voltage of 120 kV.

### 2.9. Slot Blotting

Slot blotting combined with Ponceau S protein staining was used in this study to observe the effect of DA-CDs in disaggregating preformed amyloid fibrils. A 200 μL sample from the end of ThT assay was loaded onto a Bio-Dot SF microfiltration apparatus with a vacuum system (Bio-Rad, Hercules, CA, USA). A nitrocellulose membrane with 0.2 μm pore size was used in the filtration apparatus. The remaining large protein aggregates on membrane were stained by Ponceasu S solution, which was prepared at 0.1% (*w*/*v*) in 5% acetic acid. After 1 h, the membrane was washed with DDI water several times until a clear white background was observed.

## 3. Results and Discussion

### 3.1. The Effect of CDs on hCT Fibrillization

The aggregation behavior of hCT is complex and has not been elucidated extensively. In the beginning, D_15_FNKF_19_, a fragment of hCT, was considered a crucial factor that initiates protein assembly [24]. Two Phe residues within this region have been suggested by solid-state nuclear magnetic resonance (NMR) study to stabilize β-sheets of hCT amyloid fibrils via aromatic π-π interaction [25]. Moreover, Asp15 and Lys18 could contribute possible electrostatic interactions during aggregation and they are also important in determining the orientation of molecular stacking [26]. Based on these results, we aimed to synthesize CDs with different functionality and ultrasmall sizes and then test their potentials in preventing hCT fibrillization.

The synthesis of CDs used in this study is simple and straightforward according to the reference paper [23]. We chose citric acid as a starting material for calcination because we hope CDs are passivated with carboxyl groups. It is easier to modify the surface properties of CDs by cross-linking reagents with different amine molecules, including ethanolamine, ethylenediamine, β-alanine, and dopamine (named ETA-CDs, ED-CDs, β-a-CDs, and DA-CDs, Figure 1). The hydrodynamic diameter distribution of original CDs and others measured by the dynamic light scattering (DLS) was 1–3 nm. The surface properties of CDs, as well as particle size, are listed in Table 1. The FTIR and XPS spectra of original CDs were also recorded and are shown in Appendix A. Basically, we judged whether the synthesis was successful from the change of zeta potential of CDs, and we indeed found that surface properties of original CDs were changed after coupling reactions. Later, the kinetics of hCT amyloid formation with these CDs were monitored by ThT assays. ThT is a fluorescent dye that is conventionally used in detecting amyloid fibrils formed by a variety of amyloidogenic proteins or peptides [27,28]. When it binds to β-sheet-rich structures such as those in amyloid, ThT displays enhanced fluorescence with a characteristic red shift in its emission spectrum. A previous study suggested that ThT fluorescence could correlate linearly with amyloid concentration over a range of concentrations from 0.2 to 500 μM. Therefore, we first used ThT in a series of kinetic tests to examine the inhibitory effect of CDs.

hCT aggregation was suggested to be like that of most of the amyloidogenic proteins, which may follow a nucleation-polymerization mechanism to form irreversible amyloid aggregates [29]. It has been proposed that the aggregation process may include three distinct stages: nucleation, elongation, and saturation. In the initial nucleation phase (lag phase), protein monomers associate into nucleation seeds, which is considered a rate-limiting step in the whole process. Therefore, there is a significant lag time for fibril formation, followed by a rapid elongation phase once seeds have been generated. In the elongation phase, protein oligomers continuously interact with monomers, promoting the formation of amyloid fibrils. The process of nucleation cannot be revealed by ThT assay due to the absence of fibrillar β-sheet structures. In other words, there is no detectable ThT fluorescence during lag time. The lag time for fibril formation can be dramatically reduced by the addition of preformed fibril seeds to protein monomer solutions. We chose ThT assay for monitoring hCT fibrillization because the lag time and the amounts of fibrils can be visualized by this fluorescent probe.

To find the best nanomaterials as aggregation inhibitors, we first tested the effect of CDs on hCT fibrillization. hCT monomers (64 μM) were prepared individually with five different types of CDs (200 μg/mL) in the presence of ThT (Figure 2A). We quickly noticed that one of these types of CDs, DA-CDs, would significantly inhibit hCT fibrillization due to very weak detectable fluorescence. The lag time of hCT alone examined in our condition was about 20 h. The lag time observed for hCT with the other four types of CDs was quite similar to the time measured for hCT alone, suggesting these materials did not successfully impede hCT aggregation. The absorbance and fluorescence of CDs themselves were very weak during 400~600 nm (Appendix A), and we can exclude the interference of CDs on ThT fluorescence. The inhibitory effect of DA-CDs seems to be concentration dependent (Figure 2B); ThT fluorescence became lower with the increasing concentration of DA-CDs. Here, the end products of ThT assay were further examined by TEM. A large amount of hCT fibrillar aggregates was observed in the images from hCT alone sample; however, only some dots were observed from hCT with DA-CDs (Figure 2C,D).

On the other hand, we also attempted to probe potential large aggregates during hCT aggregation by dynamic light scattering (DLS) and to see how DA-CDs modulate the production of aggregates (Figure 3). First of all, samples of hCT alone, DA-CDs alone, and hCT with DA-CDs were prepared separately in microtubes. After 48 h incubation, we noticed large particles produced from hCT alone samples and carbon dots aggregation in the solution. However, hCT in the presence of DA-CDs largely remained in soluble form. Even after 72 h, hCT alone samples have formed a very large assembly, but this was not the case for hCT with DA-CDs, suggesting DA-CDs did suppress hCT aggregation. Because of these interesting findings, we conducted more examinations for DA-CDs on hCT fibrillization.

### 3.2. DA-CDs Inhibit hCT Fibrillization in the Presence of Preformed Seeds and Lipid Vesicles

According to previous studies, the rate of hCT fibrillization is also sensitive to environmental conditions, such as solutions containing small amounts of performed fibrillary aggregates or lipids. The seeding effect is a general phenomenon observed for most of the amyloidogenic proteins. The surface structure of mature fibrils seems to catalyze the nucleation of protein monomers and accelerate fibrillogenesis. The promotion of hCT amyloid formation by phospholipid was also evident, although the mechanism was not fully elucidated [30]. Phospholipid membranes were suggested to induce the structural conversion of calcitonin and enhance its β-sheet and amyloid structure [31]. Here, we further examined the effectiveness of DA-CDs in inhibiting hCT fibrillization in the conditions that greatly facilitate peptide aggregation. We chose to add preformed hCT fibrils and lipid vesicles composed of POPC, POPS, and cholesterol (49%, 21%, and 30%) as inducing reagents. As expected, we found that the aggregation of hCT was accelerated in these two conditions including induction factors (Figure 4). Lag time was apparently eliminated in the presence of 10% preformed hCT fibrils (in monomeric units), and it was reduced to 10 h when lipid vesicles were present. Even so, DA-CDs still exhibited the inhibitory effects as shown in ThT assays. From the images collected by TEM, we only observed some atypical amorphous aggregates in the hCT samples with DA-CDs (Figure 4C,D).

### 3.3. DA-CDs Were Also Effective in Nucleation and Elongation States of Fibrillization

Next, we examined the inhibitory effect of DA-CDs during nucleation and elongation states of fibrillization. It is important because we need to know whether this material can interact with hCT oligomers before the formation of an effective nucleus or hinder monomer addition to protofibrils and the formation of new fibrils on existing fibrils. To answer this critical question, we attempted to add DA-CDs at different time points in the kinetic experiment rather than in the beginning of the reaction (Figure 5). Similar ThT results were found when DA-CDs were added at 0 h and 5 h. The final ThT intensity was much weaker in these two conditions than in hCT alone samples, but lag time did not significantly extend when DA-CDs were added at 5 h, at which time hCT were monomers and oligomers dominant. On the other hand, we also added DA-CDs at 30 h, a time point at which enhanced ThT intensity has been markedly noticed. Interestingly, the ThT intensity did not persist to go up after the addition of DA-CDs. On the contrary, the ThT signal gradually went down and remained at a similar extent as observed from the other two conditions. This interesting phenomenon may help explain the measured ThT curve when a lower concentration of DA-CDs (50 or 100 μg/mL) was applied in Figure 2B. We noticed that the ThT intensity increased after 20 h and then decreased slowly after 30–32 h, suggesting that DA-CDs preferred to interact with larger protein species such as fibrillar oligomers or protofibrils. Therefore, we speculated that DA-CDs might be capable of dissociating preformed aggregates since a difference in ThT intensity was consistently monitored from 32 h to the end of the reaction. Similar ThT curves are also observed when using flavonoids as hCT inhibitors [12].

### 3.4. The Dissociating Capability of DA-CDs Is Active and Specific to hCT Fibrils

So far, there have been several successful examples that demonstrate CDs derived from different preparations can be used to inhibit amyloid fibril formation. Most of them even focus on insulin fibrillization [32,33]. Nonetheless, the capability of CDs in dissociating preformed fibrils has not been examined before. To further confirm the previous observations in ThT studies, we first prepared mature hCT fibrils in 384-well microplates and then added DA-CDs in the reaction-plateaued state. The whole process was still monitored by collecting ThT signals. Figure 6 clearly shows that ThT intensity gradually decreased after adding DA-CDs and reached approximately half of maximum ThT signals after another 72 h. TEM images suggested that the morphology of remaining fibrils is not like that of mature fibrils. Most of them are short and loosely packed amorphous aggregates. A large reduction in density of long-bundle-like amyloid fibrils was observed when they were treated with DA-CDs. Later, we further evaluated the effect of DA-CDs in dissociating Aβ and IAPP fibrils. Aβ and IAPP monomers are similar to hCT in length. Due to the formation of a disulfide bond, IAPP also contains a cyclic loop in the N-terminus region. However, the sequence of these peptides had very little in common, even though they all can form β-sheets-enriched protein fibrils. We conducted similar approaches for Aβ and IAPP fibrils as we did for hCT. When the monitored ThT signals showed that Aβ and IAPP were in the fibrillar state, we added DA-CDs and continued to collect ThT fluorescence (Figure 6D and Appendix A). From ThT results, we found that the intensity was mostly unaffected when DA-CDs were added in Aβ and IAPP fibrillar samples, indicating those two types of protein fibrils cannot be deconstructed by DA-CDs. A minor reduction in ThT intensity observed in the Aβ case may result from dilution upon addition because a similar ThT curve was monitored in the control study. To further confirm ThT data, we monitored the sample morphologies by TEM (Figure 6B,C) and conducted a slot blot (Figure 6E) to detect large aggregates remaining in the membrane after filtration. It can be seen that the color of the band from the loading of hCT fibrils with DA-CDs is much lighter than that of the other conditions. The dissociating ability of DA-CDs seems to be specific to hCT fibrils.

### 3.5. DA-MSNs Also Can Reduce Formation of hCT Fibrils but Are Less Effective than DA-CDs

To confirm the catechol functional group of dopamine is the key feature in disrupting hCT fibrillization, we chose to conjugate dopamine on the surface of silica nanoparticles perform a similar examination as we conducted using CDs. MSNs have been widely developed in biomedical-related fields, especially in drug delivery systems, due to their unique advantages, such as easy functionality modification and large-scale production, large surface area, uniform pore size, and low cytotoxicity [34]. Although we do not need the properties of holes in our materials, we hoped the large surface area of MSN would carry more catechol moiety. The overall size of MSNs of approximately 45 nm produced in this study is much larger than that of CDs. We are not sure whether the size of materials is also a matter of their effectiveness in preventing protein aggregation. The characterizations of DA-MSNs are shown in Appendix A. The FTIR spectra of DA-MSNs largely agreed with previous data measured for MSNs or MSN-NH_2_ [35], but it cannot tell if the catechol group was successfully conjugated. The morphologies of DA-MSNs were observed by TEM, which showed that they were homogeneous spherical nanoparticles with an average diameter of about 45 nm. Here, we further measured the UV spectra for DA-MSNs and MSN-COOH processed as background signals. The characteristic UV band of dopamine at around 254 and 280 nm was shown in the spectra, indicating that the coupling of dopamine on the surface of MSN-COOH was achieved. Later, again, DA-MSNs with the same concentration as DA-CDs were examined for inhibition on hCT amyloid formation by ThT assay. The inhibitory effect of DA-MSNs was as expected and also concentration-dependent; however, DA-MSNs were much less effective than DA-CDs. The final ThT intensity of hCT fibrillization with 200 μg/mL DA-MSNs was about half of that measured for hCT alone (Figure 7). In fact, it is very difficult to rationalize why DA-CDs exhibited better inhibition from the number of catechol groups on both nanomaterials. We speculated that the stability of nanomaterials was an important factor in preventing aggregation because the dispersion of DA-CDs in aqueous solution is much better than that in DA-MSNs. The stability of DA-CDs may result from the surface charge and tiny size of particles. Although the catechol group was neutral in solution, the zeta potential of DA-CDs was not down to zero after conjugation. The negative potential charge of DA-CDs may bring additional advantages when it is used as an inhibitor.

## 4. Conclusions

In this work, we first synthesized five types of CDs and studied the effect on hCT fibrillization as primary screening. ThT fluorescence assays indicated that DA-CDs with catechol groups were effective in preventing hCT amyloid formation as validated by TEM. However, it was found that CDs with high negative potential or aliphatic hydroxyl group were not able to modulate hCT aggregation. hCT is a 32-residue short peptide composed of mostly uncharged amino acids, one aspartic acid, and one lysine. A previous solid-state ^13^C NMR study has examined the effect of the electrostatic interaction during hCT fibrillization. The results showed that the replacement of Asp15 with Asn15 did not significantly reduce the rate constants of the fibril formation. Thus, we speculated that the introduction of a negatively charged nanomaterial cannot obstruct hCT protein–protein interaction if electrostatic interaction was not the main force to promote hCT association. In addition to electrostatic interaction, aromatic–aromatic interaction was mostly acknowledged to play a critical role in the fibril formation of hCT. Polyphenolic compounds such as EGCG, Quercetin, Myricetin, and Baicalein with catechol groups in common have been revealed to effectively inhibit hCT amyloid formation. In our studies, we simply conjugated the molecules of dopamine on the surface of CDs via the formation of a covalent bond with amino groups of dopamine. Our study here strongly agrees with previous studies and more directly suggests the catechol group is a key moiety that contributes to the inhibitory effect. Moreover, we found the ThT curves are quite similar in both studies using flavonoids with vicinal hydroxyl groups and DA-CDs as inhibitors. The inhibitory effect of natural products and DA-CDs is concentration-dependent, indicated by ThT fluorescence. Moreover, when ThT intensity reached the maximum extent, it would then gradually decrease with longer incubation. We conjectured that the mature fibrils would be dissociated later, although the dissociating capability had not been tested before. Here, we confirmed that DA-CDs can disassemble hCT amyloid fibrils. Interestingly, the dissociating capability of DA-CDs is specific to hCT amyloid. Our studies suggest that this nanomaterial would be useful in hCT formulation and may also provide emerging treatments for hCT-related amyloidosis.

## Figures and Tables

**Figure 1 nanomaterials-11-02242-f001:**
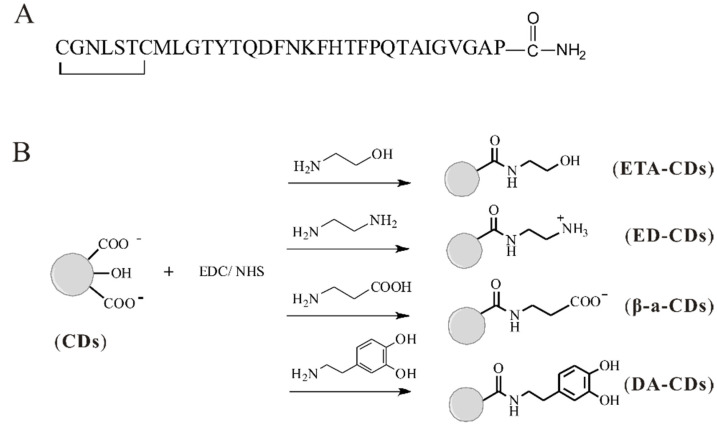
The primary sequence of hCT and schematic diagram of CDs synthesis. (**A**) hCT contains a disulfide bridge between residues 1 and 7, and the C-terminus is amidated. (**B**) A series of CDs with different functionality were obtained via modification of original CDs with different amine compounds by coupling reagents.

**Figure 2 nanomaterials-11-02242-f002:**
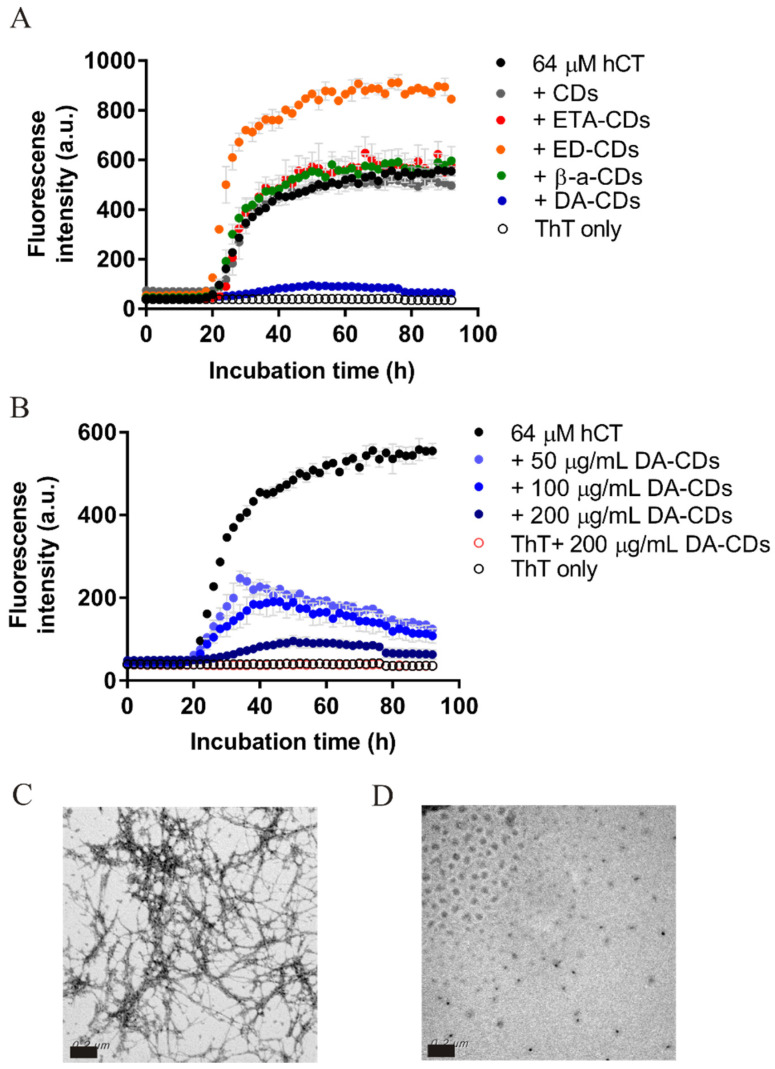
Inhibitory effect of DA-CDs on hCT fibrillization. (**A**) Kinetics of hCT amyloid formation were monitored by ThT assays. The concentration of hCT in all experiments was 64 μM, and that of CDs was 200 μg/mL. (**B**) The effects of DA-CDs on hCT fibrillization were further examined from 50 to 200 μg/mL by ThT assay. (**C**) TEM images of hCT alone sample at the end of kinetic experiments. (**D**) TEM images of hCT with 200 μg/mL DA-CDs at the end of the kinetic experiment. Scale bar represents 200 nm.

**Figure 3 nanomaterials-11-02242-f003:**
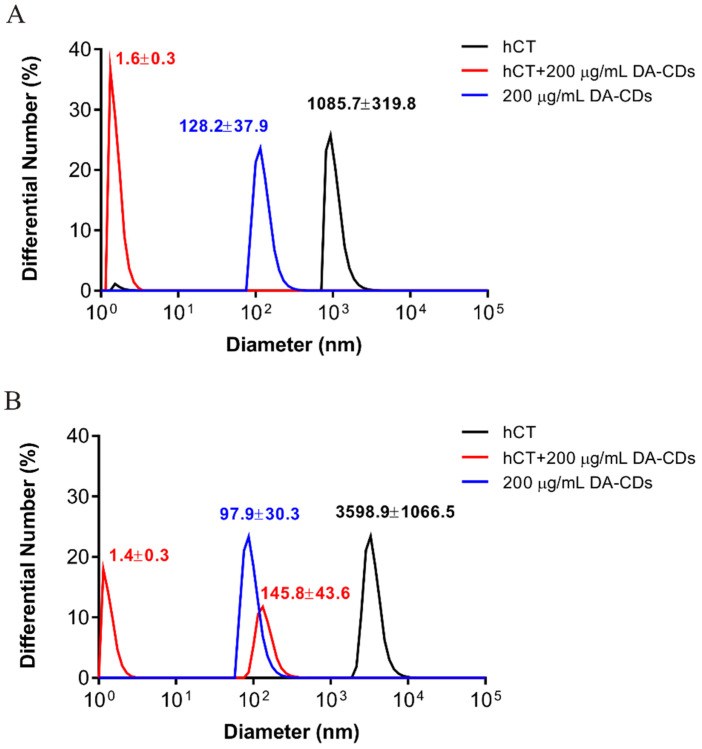
Particle distribution monitored by DLS. hCT alone (black), DA-CDs alone (blue), and hCT with DA-CDs (red) were individually incubated in microtubes and subjected to DLS at (**A**) 48 h and (**B**) 72 h.

**Figure 4 nanomaterials-11-02242-f004:**
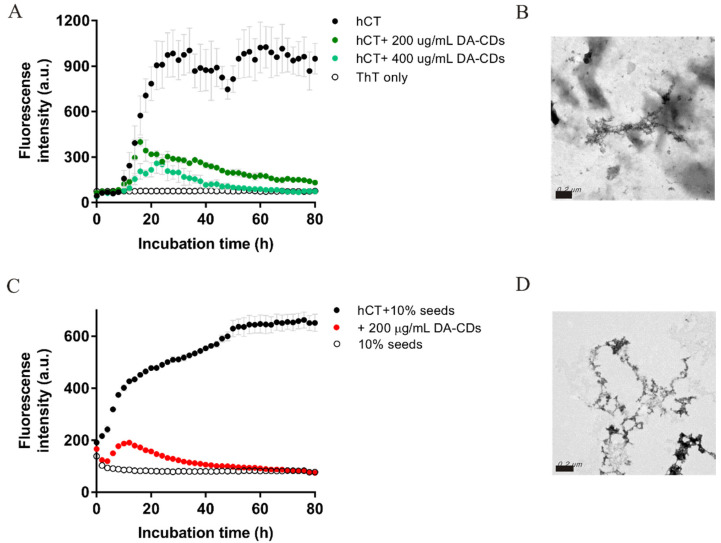
ThT kinetics monitored for hCT fibrillization in the presence of (**A**) lipid vesicles and (**B**) 10% preformed hCT amyloid fibrils used as seeds. TEM images for ThT end products with 200 μg/mL DA-CDs from conditions in the presence of (**C**) lipid vesicles and (**D**) 10% seeds. Scale bar represents 200 nm.

**Figure 5 nanomaterials-11-02242-f005:**
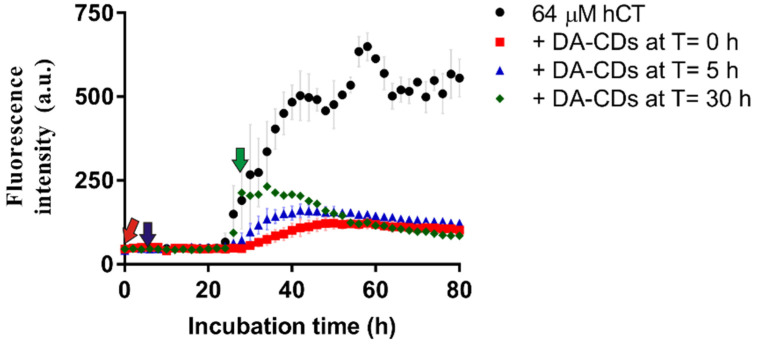
hCT aggregation kinetics monitored by ThT assay. DA-CDs were added at different time points during the aggregation process. Red arrow: 0 h; blue arrow: 5 h; green arrow: 30 h.

**Figure 6 nanomaterials-11-02242-f006:**
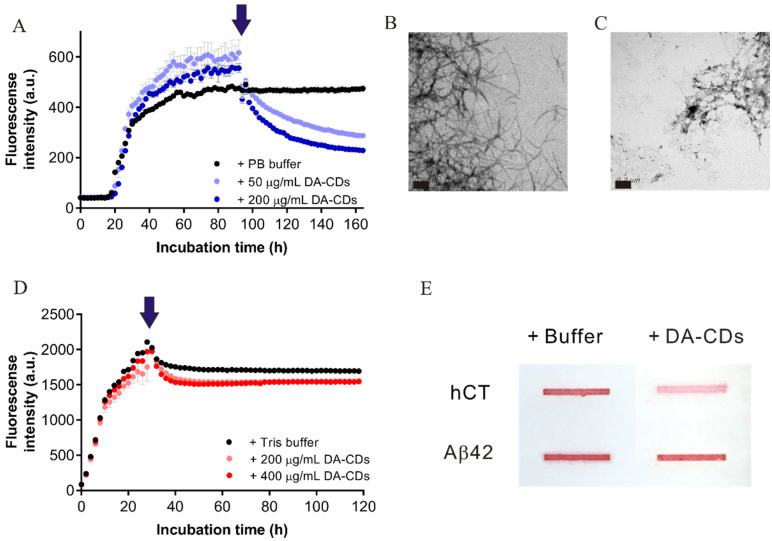
The dissociating capability of DA-CDs toward different amyloidogenic proteins. (**A**) ThT-monitored hCT fibril dissociation by DA-CDs. (**B**) TEM images for ThT end products with an addition of buffer as a control study. (**C**) TEM images for ThT end products with the addition of 200 μg/mL DA-CDs at reaction plateau. Scale bar represents 200 nm. (**D**) ThT-monitored Aβ fibril dissociation by DA-CDs. (**E**) Slot blotting for remaining large aggregates stained by Ponceau-S.

**Figure 7 nanomaterials-11-02242-f007:**
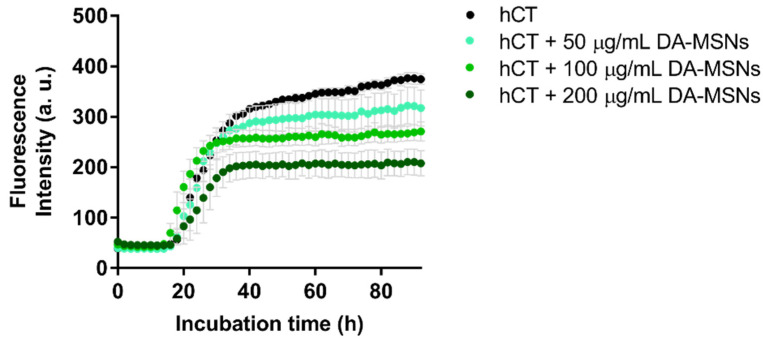
ThT-monitored hCT aggregation kinetic in the presence of DA-MSNs.

**Table 1 nanomaterials-11-02242-t001:** Mean diameter and zeta potential of five carbon dots with different functionality in DDI water.

	Diameter/nm	ζ-potential (mV)
CDs	1.5	–34.7
ETA-CDs	1.8	–0.2
ED-CDs	1.7	0.4
β-a-CDs	1.5	–24
DA-CDs	1.2	0.2

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
