# Peer review of "Dopamine-Conjugated Carbon Dots Inhibit Human Calcitonin Fibrillation"

_nanomaterials, 2021, doi:10.3390/nano11092242_

Round 1
Reviewer 1 Report
In this work, Wu and co-workers developed new carbon dots (CDs) with potential for inhibiting human calcitonin fibrillation. This is relevant topic of research with interesting therapeutic applications.
This paper appears to be globally scientifically sound (albeit with some aspects that raise some doubts), and is well written. The results show that these CDs do present significant potential, which justify publication if the authors can address the following:
-What is the synthesis yield (mass per mass, %) of the obtained CDs?
-How many replicate synthesis were performed to ensure that the obtained CDs possess reproducible properties?
-The authors should also TEM microscopy to assess the size and morphology of the obtained CDs;
-The authors used somewhat short reaction times and temperatures for the synthesis of CDs. How can the authors ensure that the obtained reaction products are indeed nanoparticles and not polymeric/oligomeric structures that result from the initial condensation reactions involving citric acid, which occur in bottom-up routes? In fact, the weak absorbance/fluorescence of CDs could support my hypothesis;
-It is not clear to me if, in the ThT assays, if the authors checked the fluorescence of ThT just in the presence of CDs, and in the absence of hCT monomers. This could be useful to ensure that CDs do not affect directly the fluorescence of ThT;
-It would be useful if the authors performed in vitro cytotoxicity assays involving DA-CDs, in order to ensure that these CDs are biocompatible and useful in the future for in vivo therapeutic approaches.
Author Response
Point 1: What is the synthesis yield (mass per mass, %) of the obtained CDs?
Response 1: The synthesis yield of citric acid-derived CDs after dialysis is about 15%. We have added this information in method 2.2.
Point 2: How many replicate synthesis were performed to ensure that the obtained CDs possess reproducible properties?
Response 2: The CDs were synthesized three times after we confirm the reference protocol. The size and zeta potential of CDs were measured every time to ensure that the properties are similar between batches. DA-CDs were synthesized twice to ensure their inhibition in hCT fibrillization. We also added this information in method 2.2.
Point 3: The authors should also TEM microscopy to assess the size and morphology of the obtained CDs
Response 3: We did not typically take a TEM image for CDs only. However, we can see those dots images from the end products of ThT assay for hCT with DA-CDs (Figure 2D) demonstrating that we did add DA-CDs in the reaction.
Point 4: The authors used somewhat short reaction times and temperatures for the synthesis of CDs. How can the authors ensure that the obtained reaction products are indeed nanoparticles and not polymeric/oligomeric structures that result from the initial condensation reactions involving citric acid, which occur in bottom-up routes? In fact, the weak absorbance/fluorescence of CDs could support my hypothesis
Response 4: From points of our view, it is difficult to well define CDs. However, in general, CDs were considered also encompass graphene quantum dots and polymer dots. CDs should consist of sp2/sp3 carbons with some connected functional groups with tunable emission optical properties and are less than 10 nm in size. First of all, we search for many bottom-up synthetic protocols to afford CDs. We follow the method (calcination time and temperature) suggested from reference 18 (Chem. Commun., 2014, 50, 7318) because thermogravimetric analysis and biocompatibility of CDs have been well characterized. Later, dialysis was performed after calcination using 1 kDa dialysis bag to remove any potential small or oligomeric molecules. We confirmed that the obtained CDs synthesized in our lab with tunable emission but weak fluorescence. In fact, weak fluorescence is also one of our considerations for CDs preparation. We don’t want the strong fluorescence of CDs to interfere ThT fluorescence assay. The same protocol would allow us to make very bright CDs if some amine small molecules (ethylenediamine) were added to citric acids.
Point 5: It is not clear to me if, in the ThT assays, if the authors checked the fluorescence of ThT just in the presence of CDs, and in the absence of hCT monomers. This could be useful to ensure that CDs do not affect directly the fluorescence of ThT.
Response 5: We did check the ThT fluorescence in the presence of CDs without hCT during monitored aggregation process. We would like to add this data to Figure 2B.
Point 6: It would be useful if the authors performed in vitro cytotoxicity assays involving DA-CDs, in order to ensure that these CDs are biocompatible and useful in the future for in vivo therapeutic approaches.
Response 6: We thank the review’s suggestion. We will perform a cytotoxicity assay for CDs with hCT in the following studies. We did not test cytotoxicity for CDs and DA-CDs here because they have been evaluated in reference and the concentration used in our study was to a similar extent in references studies.

Reviewer 2 Report
This work report an interesting application of conjugated carbon dots in inhibiting fibril formation. The research is well designed and the results are well presented. However, it is critical to check the biocompatibility of the dopamine conjugated carbon dots and determine the safety concentration level that can be applied.
Author Response
Point 1: This work report an interesting application of conjugated carbon dots in inhibiting fibril formation. The research is well designed and the results are well presented. However, it is critical to check the biocompatibility of the dopamine conjugated carbon dots and determine the safety concentration level that can be applied.
Response 1: We thank the review’s suggestion. We will perform a cytotoxicity assay for CDs with hCT in the following studies. We did not test cytotoxicity for CDs and DA-CDs here because they have been evaluated in reference and the concentration used in our study was to a similar extent in references studies (Chem. Commun., 2014, 50, 7318; ACS Appl. Mater. Interfaces 2015, 7, 42, 23564–23574)

Reviewer 3 Report
In this manuscript, the author reports, ‘Dopamine conjugated carbon dots inhibit human calcitonin fibrillation’. The current study is on a topic of relevance and general interest to readers in this area. The authors should address the following questions before getting a possible publication.
Recommendation: Major revisions needed as noted.
- The scale bars in Figure 2C, Figure 2D, Figure 4B, Figure 4D, Figure 6B, Figure 6C are not visible.
- The novelty of the present work should be discussed in the Introduction section.
- The author should write the purpose for each test in one/two sentences (in brief) before explaining the results of the characterization techniques. Therefore, the logic and organization of this part will be enhanced.
- What about photostability of the CDs? The authors are encouraged to perform some photostability experiments of the CDs.
- The formatting and grammatical errors in the article need to be checked carefully.
- The authors are encouraged to provide one comparison table of the CDs synthesis, sizes, optical properties, and applications with the reported studies.
- The authors have cited relevant references in the Introduction section; however the topic background of the manuscript needs to be lighted further to broaden the impact, related literatures: Analytica Chimica Acta, 995, 99-105; ACS Applied Materials & Interfaces, 13(26), 31038-31050; Acta Biomaterialia 78 (2018): 178-188; ACS Applied Nano Materials, 3(12), 11777-11790
Author Response
Point 1: The scale bars in Figure 2C, Figure 2D, Figure 4B, Figure 4D, Figure 6B, Figure 6C are not visible.
Response 1: We have enlarged the scale bars to make them clear to see.
Point 2: The novelty of the present work should be discussed in the Introduction section.
Response 2: We thank the reviewer’s suggestion. We have added more sentences (page 3.) to describe the novelty of this work.
Point 3: The author should write the purpose for each test in one/two sentences (in brief) before explaining the results of the characterization techniques. Therefore, the logic and organization of this part will be enhanced.
Response 3: We thank the reviewer’s suggestion. We have checked if there are appropriate descriptions to show the purpose of each test in each paragraph of Results and Discussion. And, we have added some sentences if they are missing in the first version.
Point 4: What about photostability of the CDs? referring to ACS Applied Nano Materials, 3(12), 11777-11790; Colloids and Surfaces A: Physicochemical and Engineering Aspects 579 (2019): 123604
Response 4: We did not test photostability for our obtained CDs because they did not exhibit optical properties with strong fluorescence. Most of the CDs with strong fluorescence are usually synthesized with amine compounds (N-doped carbon dots) and can be used as a chemical sensor. We did not need this optical property in preventing protein aggregation. Thus, we only used citric acid as precursor chemicals. Weak fluorescence is also one of our considerations for CDs preparation. We don’t want the strong fluorescence of CDs to interfere ThT fluorescence assay.
Point 5: The formatting and grammatical errors in the article need to be checked carefully.
Response 5: We have checked the entire article again to avoid grammatical errors.
Point 6: The authors are encouraged to provide one comparison table of the CDs synthesis, sizes, optical properties, and applications with the reported studies.
Response 6: We thank the reviewer’s suggestion. However, the novelty of this study is not on the preparation of CDs but its application in protein chemistry. We followed the reference paper (Chem. Commun., 2014, 50, 7318) to afford CDs and are the first group to use CDs in hCT studies. So far, there are no similar reported studies.
Point 7: The authors have cited relevant references in the Introduction section; however the topic background of the manuscript needs to be lighted further to broaden the impact, related literatures: Analytica Chimica Acta, 995, 99-105; ACS Applied Materials & Interfaces, 13(26), 31038-31050; Acta Biomaterialia 78 (2018): 178-188; Materials Science and Engineering: C 75 (2017): 1456-1464
Response 7: We thank the reviewer for providing excellent suggestions. We have mentioned a couple of different CDs and their applications. Hope that will convince more reader the advantage of using CDs.

Round 2
Reviewer 1 Report
The authors have addressed my comments, and so, my recommendation is for acceptance.
Reviewer 3 Report
The authors have addressed all the questions raised before. Therefore, the manuscript can be accepted now.